# Epidemiology of Hand, Foot, and Mouth Disease and Genetic Evolutionary Characteristics of Coxsackievirus A10 in Taiyuan City, Shanxi Province from 2016 to 2020

**DOI:** 10.3390/v15030694

**Published:** 2023-03-07

**Authors:** Jitao Wang, Hongyan Liu, Zijun Cao, Jihong Xu, Jiane Guo, Lifeng Zhao, Rui Wang, Yang Xu, Ruihong Gao, Li Gao, Zhihong Zuo, Jinbo Xiao, Huanhuan Lu, Yong Zhang

**Affiliations:** 1School of Public Health, Shanxi Medical University, 56 Xinjian South Road, Taiyuan 030001, China; 2Taiyuan Center for Disease Control and Prevention, 89 Xinjian South Road, Taiyuan 030012, China; 3WHO WPRO Regional Polio Reference Laboratory, National Health Commission Key Laboratory of Biosafety, National Health Commission Key Laboratory of Medical Virology, National Institute for Viral Disease Control and Prevention, Chinese Center for Disease Control and Prevention, Beijing 102206, China; 4Center for Biosafety Mega-Science, Chinese Academy of Sciences, Wuhan 430071, China

**Keywords:** hand, foot, and mouth disease, coxsackievirus A10, RT-PCR, *VP1* region, phylogenetic tree

## Abstract

In recent years, the prevalence of hand, foot, and mouth disease (HFMD) caused by enteroviruses other than enterovirus A71 (EV-A71) and coxsackievirus A16 (CVA16) has gradually increased. The throat swab specimens of 2701 HFMD cases were tested, the *VP1* regions of CVA10 RNA were amplified using RT-PCR, and phylogenetic analysis of CVA10 was performed. Children aged 1–5 years accounted for the majority (81.65%) and boys were more than girls. The positivity rates of EV-A71, CVA16, and other EVs were 15.22% (219/1439), 28.77% (414/1439), and 56.01% (806/1439), respectively. CVA10 is one of the important viruses of other EVs. A total of 52 CVA10 strains were used for phylogenetic analysis based on the *VP1* region, 31 were from this study, and 21 were downloaded from GenBank. All CVA10 sequences could be assigned to seven genotypes (A, B, C, D, E, F, and G), and genotype C was further divided into C1 and C2 subtypes, only one belonged to subtype C1 and the remaining 30 belonged to C2 in this study. This study emphasized the importance of strengthening the surveillance of HFMD to understand the mechanisms of pathogen variation and evolution, and to provide a scientific basis for HFMD prevention, control, and vaccine development.

## 1. Background

Hand, foot, and mouth disease (HFMD) is a global, acute febrile exanthematous infectious disease usually caused by human enterovirus (EV) and occurs mainly in young children less than 5 years of age [1]. HFMD is a mild disease but sometimes can cause severe complications that can even lead to death [2,3]. EV belongs to the genus *Enterovirus*, family *Picornaviridae*, order *Picornavirales.* EV is highly infectious to susceptible populations, and EVs that infect human beings can be divided into four species (A–D) [4]. EV-A consists of 25 viruses such as coxsackievirus A2 (CVA2)-CVA8, CVA10, CVA12, CVA14, CVA16, EV-A71, EV-A76, EV-A89-A92, EV-A114, and EV-A119-A125, that can cause HFMD, acute flaccid paralysis (AFP), herpetic angina, encephalitis meningitis, and other diseases. HFMD is one of the most common diseases caused by the members of EV-A [5,6]. Although EV-A71 and CVA16 were mainly responsible for causing HFMD [7,8,9], the recent increased frequency of CVA10-associated HFMD outbreaks worldwide, have changed this view [10,11]. In fact, the pathogen spectrum of HFMD continues to change, CVA10 even replaced EV-A71 and CVA16 as the dominant pathogen for HFMD in some areas. CVA10-associated HFMD outbreaks have been reported in past decades, in several countries including France, Vietnam, Uruguay [12,13,14], as well as in Fujian Province, Wuhan City, and Shijiazhuang City in China [15,16,17,18]. However, the CVA10 genome is less stable than that of EV-A71, CVA16, and CVA6, with more frequent recombination [19]. HFMD associated with CVA10 is more severe, and more likely to cause serious nervous system diseases, such as aseptic meningitis and viral meningitis, especially in younger children, and is the new leading cause of severe HFMD in addition to EV-A71 [10,19]. Thus, more research on the genetic characteristics and evolution of the CVA10 epidemic is required.

The enterovirus coding region is divided into three subregions, P1, P2, and P3. P1 encodes four structural proteins (VP1, VP2, VP3, and VP4), whilst the non-structural proteins are encoded in the P2 (2A, 2B, and 2C) and P3 (3A, 3B, 3C, and 3D polymerase) regions. VP1, VP2, and VP3 are present on the outside of the capsid, whereas VP4 is internal. The VP1 protein of CVA10 can identify and bind to receptors on target cells, which contains many important neutralization epitopes [20,21]. Phylogenetic analysis based on the nucleotide sequence of the entire *VP1* region of CVA10 (894 bp) is widely used to understand its genetic evolutionary characteristics and is a reliable and rigorous method for molecular epidemiological studies [18,22].

Taiyuan city, the capital of Shanxi Province, China, is located to the west of Taihang Mountain and the east of the Yellow River, on the Loess Plateau in Northwestern China. CVA10-associated HFMD had been reported in many provinces and regions of China, but only a few studies were published from Taiyuan City. In this study, we analyzed the epidemiological characteristics of HFMD and the genetic characteristics of CVA10 in Taiyuan City, Shanxi Province, from 2016–2020. The study emphasized the importance of strengthening the surveillance of HFMD to understand the mechanisms of pathogen variation and evolution, so as to provide a scientific basis for HFMD prevention, control, and vaccine development. 

## 2. Methods

### 2.1. Epidemiological Information and Sample Collection

The epidemiological data were retrieved from the China Information System for Disease Control and Prevention and the population data came from the Taiyuan Statistical Yearbook. In ten hospitals in different districts and counties of Taiyuan City, specimens were collected from clinically diagnosed HFMD cases within three days after the onset of the disease and sent to Taiyuan Center for Disease Control and Prevention (Taiyuan CDC). At least five specimens were collected every month. Due to the impact of COVID-19, the number of HFMD cases and specimens in 2020 is limited. A total of 2701 throat swab specimens were sent to the virus microbiology laboratory of Taiyuan CDC for enterovirus nucleic acid testing within 24 h from 1 January 2016 to 31 December 2020. Upon receipt of specimens by the laboratory, each specimen was assigned a unique laboratory code and entered into the Gastrointestinal Information Database of Taiyuan CDC (Excel).

### 2.2. RNA Extraction and EV Identification

Viral RNA was extracted from the throat swab specimens using a MagMAX^TM^-96 Viral RNA Isolation Kit (Thermo Fisher Scientific, Foster City, CA, USA) according to the manufacturer′s instructions. RNA was eluted in a final volume of 50 μL of elution buffer and used immediately or stored at −80 °C. All steps of specimen preparation and RNA extraction were carried out in the biosafety cabinet. RNA was detected using three-channel real-time RT-PCR kits (Diagnostic Kit for pan-EV, EV-A71, CVA16, Shanghai ZJ Bio-Tech Co., Ltd., Shanghai, China). A 25 μL reaction system consisting of 1 μL enzyme mixture,19 μL reaction solution, and 5 μL viral nucleic acid was prepared. PCR cycling parameters were set up as per the manufacturer’s protocols: 50 °C for 30 min, 95 °C for 10 min, followed by 45 cycles of 95 °C for 10 s, and 55 °C for 40 s in a CFX96 Real-time Thermal Cycler (Bio-Rad, Hercules, CA, USA). A positive result was defined as a cycle threshold (Ct) value ≤ 43, and the positive control was defined as a Ct value ≤ 35. The negative specimens were not under the scope of our study. The specimens positive for other EVs were detected using commercially available real-time RT-PCR kits (Nucleic Acid Detection Kit for CVA10, BioMax Biotechnology Co., Ltd., Beijing, China). A 20 μL reaction system consisting of a 5 μL primer probe mixture, 5 μL RT-PCR reaction solution (containing enzymes), and 10 μL viral nucleic acid was prepared. PCR cycling parameters were set up according to the instructions: 25 °C for 2 min, 53 °C for 10 min, 95 °C for 2 min, followed by 45 cycles of 95 °C for 5 s, and 60 °C for 30 s in a CFX96 Real-time Thermal Cycler (Bio-Rad, Hercules, CA, USA). A positive result was defined as a Ct value ≤ 37, and the positive control was defined as a Ct value ≤ 32. The negative specimens were not under the scope of our study.

### 2.3. VP1 Region Amplification and Nucleotide Sequencing

In total, 31 specimens were detected to amplify the entire *VP1* region of CVA10 by using the One Step RT-PCR kits (CVA10 Target region (*VP1*) Molecular typing Kit, BioMax Biotechnology Co., Ltd., Beijing, China) with specific in-house primer pairs (forward primer with nucleotide position 2172–2192nt, CVA10-VP1-S: GCTCAGTAACACTCAYTTYCG, and reverse primer with nucleotide position 3376–3394nt, CVA10-VP1-A: CTCGAGAACTGTCYTCCCA). The primers yielded a CVA10 amplification product of 1223 bp, spanning the entire *VP1* region of CVA10 (894 bp). The reaction system for each tube consisted of 12.5 μL PCR Buffer, 2 μL PCR Enzyme, 0.5 μL RT Enzyme, 3μL Primer Mix, and 7 μL viral nucleic acid up to a final volume of 25 μL. The amplification conditions were as follows: 45 °C for 10 min, 94 °C for 2 min, followed by 45 cycles of 98 °C for 10 s, 55 °C for 15 s, and 68 °C for 20 s in an ABI ProFlex™ 96-Well PCR System (Thermo Fisher Scientific, Foster City, CA, USA). Amplification products were determined using a 1% agarose gel electrophoresis image analysis system and sent to Tsingke (Beijing, China) Biotechnology Co., Ltd. for DNA bi-directional sequencing.

### 2.4. Phylogenetic Analyses

Sequences were spliced by Sequencher software (version 5.4.6), and EV-types were verified by the enterovirus automated genotyping tool [23]. BioEdit7.2.5 was used for amino acid sequence alignment of the CV-A10 VP1 region. Phylogenetic trees were constructed based on the *VP1* sequences of the CVA10 identified in this study and the sequences downloaded from the GenBank database using the MEGA software (Version 11.0.11) [24]. Phylogenetic analysis using the neighbor-joining (NJ) method was performed based on the Kimura 2-parameter model, and the reliability was evaluated by 1000 bootstrap replicates. Bootstrap values greater than 80% were considered statistically significant for grouping. Genotype differences were computed by the group mean distance computing method in MEGA software.

### 2.5. Nucleotide Accession Number

All the CVA10-Taiyuan sequences obtained during this study were submitted in GenBank under accession numbers OP244619—OP244649, which are shown in Appendix A.

### 2.6. Statistical Analyses

Data analyses were performed using IBM SPSS Statistics software (Version 26.0). The Pearson’s Chi-Square Test was used to analyze the data, and the value of *p* < 0.05 was considered statistically significant.

## 3. Results

### 3.1. Epidemiology Features

A total of 22,044 clinical HFMD cases were reported to the China Information System for Disease Control and Prevention in Taiyuan during 2016–2020. The average annual HFMD incidence rate was 86.45 per 100,000 population (range 20.68–148.49), and the incidence rate increased from 2016 to 2017 and decreased from 2017 to 2020, with a peak in 2017. Among these HFMD cases, 13,152 were males and 8892 were females, with an average annual male-to-female ratio of 1.48 (range 1.36–1.56). The average annual incidence in males (101.10 per 100,000 population) was higher than in females (71.18 per 100,000 population) and this difference was statistically significant (χ^2^ = 660.330, *p <* 0.01). Of all cases, 81.65% (17,998) were children aged 1–5, and the average annual incidence is higher in boys than in girls. The majority (99.35%) of them were scattered children (49.33%), kindergarten children (41.74%), and school-going students (8.27%) (Table 1). Although HFMD cases occurred throughout the year, two seasonal peaks were observed from June to August and from October to November during 2016–2019, respectively. The annual HFMD cases were lowest (n = 1100) in 2020 due to the impact of the COVID-19 pandemic, with the peak in November (Figure 1). There is no relationship between the age and serotype of viruses according to the existing data of statistical analysis (*p* < 0.05).

### 3.2. Laboratory Detection

A total of 2701 throat swab specimens were collected from HFMD patients during 2016–2020, among them, 1439 (53.28%) were positive for human enterovirus (EV-positive), with 219 (15.22%) were associated with EV-A71, 414 (28.77%) with CVA16, and 806 (56.01%) with other EVs other than EV-A71 and CVA16. The proportion of other EVs was significantly higher than that of EV-A71 and CVA16 in Taiyuan during 2016–2020. CVA10 (31 cases) accounted for 2.15% of all confirmed cases (Table 1).

### 3.3. Phylogenetic Analysis of the VP1 Region of CVA10

The entire *VP1* sequences of 31 CVA10 were amplified and sequenced in our study. The *VP1* nucleotide sequences among CVA10 strains shared 92.6% to 100% nucleotide identity, corresponding to a 96.9% to 100% amino acid identity. A similar analysis with the prototype strain Kowalik showed a 75.9% to 77.0% nucleotide identity, corresponding to a 91.2% to 93.6% amino acid identity. The highest nucleotide similarity with the genotype C downloaded from the GenBank was 91.7% to 100% and amino acid similarity of 96.9% to 100%. A total of 52 CVA10 strains were used for phylogenetic analysis for the *VP1* region, including the 31 CVA10 strains identified in this study and 21 reference strains of CVA10 downloaded from GenBank.

The mean distance of nucleotides and amino acid sequences within and between seven genotypes of CVA10 was calculated using MEGA 11.0.11 with a p-distance model (Table 2). A difference of at least 15% in the *VP1* region was used to discriminate genotypes [25]. The results showed that the mean nucleotide distance between genotypes was more than 15%, and the mean nucleotide distance within genotypes was 3.1–8.6%. The average nucleotide divergences between genotype A and genotype G is the largest (30.5%), the difference between genotype C and genotype G is the smallest (17.5%), and the difference between other genotypes is between 18.3% and 29.9%.

As entire *VP1* sequences of CVA10 available in the public database were limited, to investigate the origin and spread of CVA10 strains in the world, three representative strains of genotype F on partial *VP1* were selected to perform the phylogenetic analysis. Seven genotypes (A, B, C, D, E, F, and G) were assigned with a mean group distance of 17.5–30.5%, and the distribution of these genotypes showed geographic and temporal circumscribed characterization. Genotype A was only composed of the prototype strain Kowalik isolated in the US in 1950. Foreign epidemic strains such as African viruses, European viruses, and Indian viruses are mainly clustered in genotypes D, E, and F. According to continuous HFMD surveillance, genotype B circulating mainly in China completely disappeared during 2004–2009 [18].

Thirty-one Taiyuan CVA10 strains along with the domestic reference strains obtained on GenBank belonged to C1 (one strain) and C2 subtypes (30 strains) within genotype C. The only CVA10 belonging to subtype C1 was acquired in 2018. Subtype C2 can be further divided into three evolutionary branches (C2a, C2b, and C2c) (Figure 2). An apparent evolutionary trend could be observed from 2016–2020. All the CVA10 strains detected in 2016 belonged to C2a. One CVA10 strain detected in 2018 and two CVA10 strains detected in 2019 belonged to C2b, whereas the remaining CVA10 strains detected between 2018 and 2020 belonged to C2c.

### 3.4. Antigenic Differences among Different Genotypes of CVA10

The 162–176 residue of CVA10 VP1 shows the effective neutralization of the CVA10 type. It is in the EF loop of VP1 protein and is a specific linear neutralizing epitope on CVA10 VP1 [26]. The amino acid sequence analysis of CVA10 genotypes including Taiyuan CVA10 showed that the linear neutralizing epitope was highly conservative in CVA10 (Figure 3).

## 4. Discussion

The HFMD epidemic has become a serious public health threat and economic burden in the Asia-Pacific region over the past decade [27,28]. The Health Ministry of China described the disease as a notifiable infectious disease of class C and established a national enhanced surveillance system for HFMD in May 2008 [2]. According to the data recorded by the China Disease Control and Prevention Information System at the end of 2021, the number of HFMD cases in Shanxi Province ranked among the top three in class C notifiable diseases. Nonetheless, studies focused mostly on the EV-A71 and CVA16 and information about other EVs was still limited [29,30]. In December 2015, China’s Food and Drug Administration approved the monovalent EV-A71 vaccine as an efficient and economic measure to control EV-caused diseases [31]. It approved the application of the monovalent EV-A71 vaccine in 2016 and has been used in Taiyuan City, Shanxi Province since 2017. Children ages 6–71 months are voluntarily vaccinated with the EV-A71 vaccine at their own expense. The annual vaccination rate of children ages 6–71 months is 15.0% (2018) to 20.8% (2021). The vaccine performed excellently in the prevention and control of EV-A71 during the past four years and changed the pathogen spectrum of HFMD. The number of HFMD cases caused by other EVs has gradually increased, even replacing EV-A71 and CVA16 as the dominant pathogen for HFMD. CVA10-associated HFMD outbreaks were reported in many provinces of China, gradually becoming one of the most frequent EV genotypes during the epidemic interval of EV-A71 and CVA16 since 2012 [18]. According to the continuous surveillance of HFMD in China, the genotype B of CVA10 prevalent in China from 2004 to 2009 has completely disappeared. The CVA10 epidemic in various provinces of China has belonged to genotype C, in recent years. Although genotype C of CVA10 belongs to C1 and C2 subgenotypes (including C2a, C2b, and C2c evolution branches), there is no difference in epidemiology. However, there are currently no preventive vaccines and specific therapeutic drugs for CVA10-associated HFMD, only a few studies were published from Taiyuan City. Hence, in this study, we conducted a comprehensive analysis of the epidemiological characteristics and molecular evolution characteristics of CVA10 in Taiyuan City from 2016 to 2020.

The results indicated that more infections were found in males than in females. Except for sex differences at the level of host immune status, another explanation could be related to the extra outdoor activities, including more frequent exposure to unclean toys and facilities by boys, which increases the risk of infection compared to girls [32]. Of all HFMD cases in Taiyuan City, children aged 1–5 accounted for the majority (81.65%) of the cases due to low herd immunity, and the majority (99.35%) among them were scattered children, kindergarten children, and school students. Such high incidence rates suggest that this particular age group should be used as a key population for HFMD control and prevention. Health education programs should be conducted to prevent HFMD transmission not only for children at kindergartens and schools but also for community families with young children. There were obvious seasonal features in the HFMD epidemic. Although HFMD cases in Taiyuan City were reported throughout the year, two seasonal peaks were observed from June to August and from October to November during 2016–2019, respectively. The main peak might be due to an increase in viral transmission because of extremes temperature [33]. The second peak may be due to a shift in climate from warm dry conditions to cold wet conditions, mainly in the southern provinces. However, this phenomenon is relatively rare in Shanxi, a northern province of China. Nevertheless, epidemiological surveillance and prevention efforts should be strengthened before the peak of the disease each year.

As one of the important viruses of other EVs, the entire *VP1* sequences of 31 CVA10 were amplified and sequenced from 2016–2020. CVA10 strains in this study belonged to C1 and C2 subtypes in genotype C and were also the predominant genotype during 2010–2016 in China [18]. Only one CVA10 isolated in 2018 belonged to subtype C1 and the others to subtype C2. Subtype C1 in our study was closely related to other domestic strains, such as Beijing, Guangxi, and Liaoning [34]. As the spread of subtype C2, it has been divided into 3 branches (C2a, C2b, and C2c). Significant branch shifts can be observed from 2016 to 2020, C2b and C2c co-circulate in 2018 and which branch will become mainstream requires our continuous surveillance.

According to the mean distance of nucleotides and amino acid sequences within and between seven genotypes of CVA10, it was found that the mean nucleotide distance within the genotypes was 3.1–7.3% except for genotype F (8.6%), which might be due to the available partial *VP1* of genotype F, it is necessary to conduct continuous surveillance to get the complete *VP1* gene. The nucleotide identity between subtype C1 and C2 was 92.6% to 100%, while the amino acid identity was 96.9% to 100%, and the nucleotide identity within subtype C2 was 94.4% to 100%, while the amino acid identity was 97.6% to 100%, showing that synonymous mutations existed generally.

Previous studies have shown that the 162–176 residues in the VP1 protein of CVA10 are a specific linear neutralizing epitope. Neutralizing epitopes play a key role in vaccine effectiveness and can be used as biomarkers to monitor vaccine effectiveness [35]. The amino acid sequence analysis of CVA10 genotypes including Taiyuan CVA10 showed that the linear neutralizing epitope (162–176 residues in VP1) was highly conservative, which might be beneficial to the development of the CVA10 vaccine. However, thus far, the research on the CVA10 epitopes is still very insufficient. One of the future research focuses is still the research on the CVA10 epitopes in order to strengthen the monitoring of the variation of the epitopes, which is also very important for the research of the CVA10 vaccine.

This study had some limitations. First, the proportion of laboratory-confirmed HFMD cases was small due to the limited sample size. Second, because there is no complete *VP1* sequence of genotype F in GenBank, partial *VP1* region sequences of genotype F of CVA10 were used for analysis in this study. However, to fully understand the genetic variation and evolution of CVA10, research based on the entire *VP1* region or even a full-length genome is necessary.

## 5. Conclusions

The study emphasized the importance of strengthening the surveillance of the HFMD pathogen spectrum in order to understand the mechanisms of pathogen variation and evolution, so as to provide a scientific basis for HFMD prevention, control, and vaccine development. The reported HFMD cases associated with CVA10 in this study raise concerns about the role of CVA10 in explaining new HFMD cases and outbreaks and call for enhanced surveillance of the HFMD pathogen spectrum and surveillance of enterovirus etiology.

## Figures and Tables

**Figure 1 viruses-15-00694-f001:**
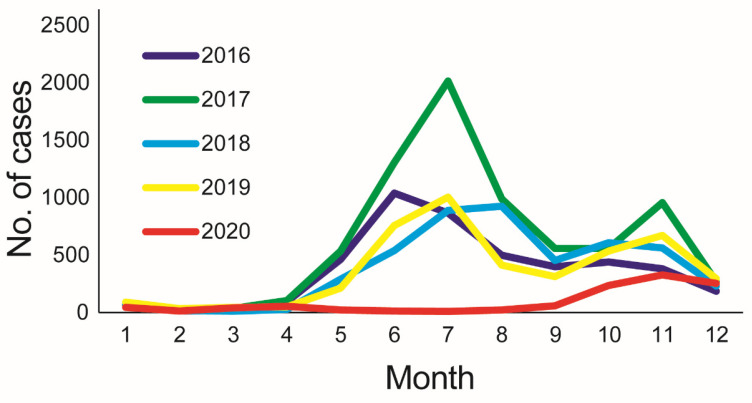
Monthly distribution of HFMD cases from 2016–2020.

**Figure 2 viruses-15-00694-f002:**
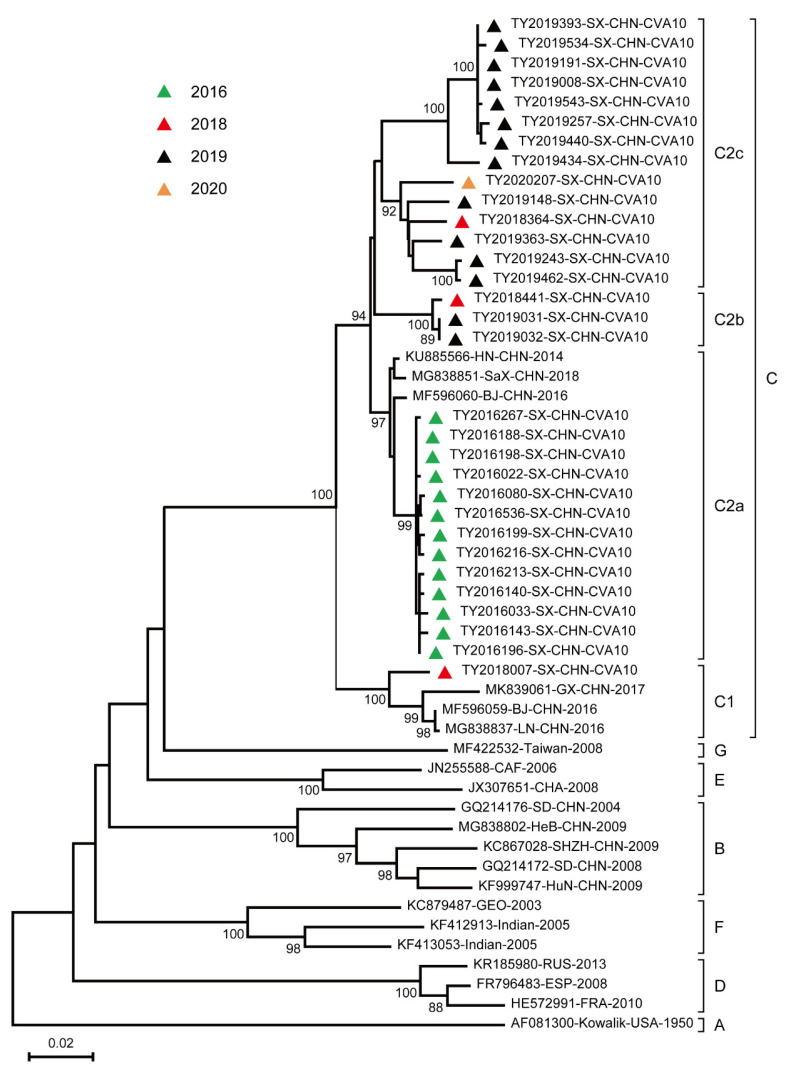
Phylogenetic analysis based on the complete and the available partial VP1 of CVA10. Taiyuan strains of different years were revealed by different colors according to the figure. The CVA10 nucleotide sequences in this study are represented by Serial number-Province-Country-Genotype.

**Figure 3 viruses-15-00694-f003:**
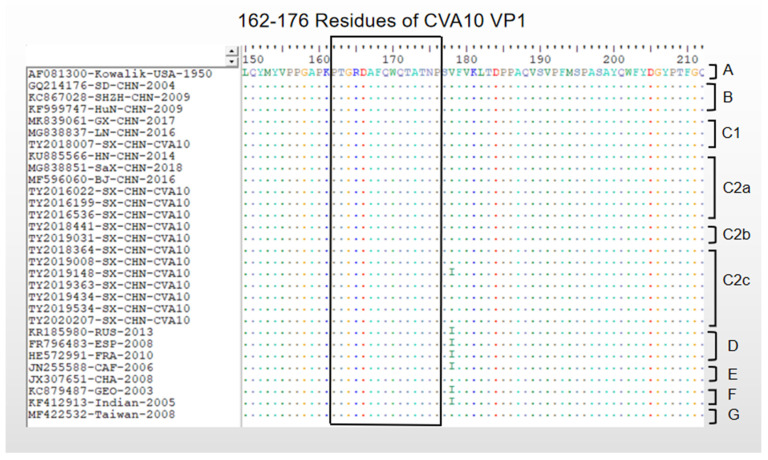
Alignment of 162–176 residues of VP1 amino acid sequences of CV-A10. According to the phylogenetic analysis of CV-A10 (Figure 2), the VP1 amino acid sequences of 30 CV-A10 in genotype A-G were selected. Dots represent the same residues as the prototype strains (AF081300-Kowalik-USA-1950). The regions of 162–176 residues are framed with black lines.

**Table 1 viruses-15-00694-t001:** Distribution of reported and detected HFMD cases in Taiyuan from 2016 to 2020.

Variable	2016	2017	2018	2019	2020	Total
HFMD Reported Cases	4472	7400	4648	4424	1100	22,044
Incidence rate/10^5^	92.17	148.49	90.87	84.56	20.68	86.45
Gender						
Male (Incidence rate/10^5^)	2728 (110.15)	4395 (172.94)	2723 (104.72)	2673 (100.44)	633 (23.19)	13,152 (101.10)
Female (Incidence rate/10^5^)	1744 (73.43)	3005 (123.04)	1925 (76.55)	1751 (68.12)	467 (18.04)	8892 (71.18)
Gender ratio	1.56	1.46	1.41	1.53	1.36	1.48
Residence						
Scattered children	2290	3691	2459	1896	539	10,875
Kindergarten children	1839	3204	1722	2012	425	9202
School students	316	458	443	489	118	1824
Other	27	47	24	27	18	143
Age Group (%)						
<1	197 (4.41%)	572 (7.73%)	291 (6.26%)	188 (4.25%)	65 (5.91%)	1313 (5.71%)
1–<2	915 (20.46%)	1655 (22.36%)	1135 (24.42%)	786 (17.77%)	204 (18.55%)	4695 (20.71%)
2–<3	810 (18.11%)	1060 (14.32%)	742 (15.96%)	693 (15.66%)	162 (14.73%)	3467 (15.76%)
3–<4	893 (19.97%)	1620 (21.89%)	760 (16.35%)	910 (20.57%)	244 (22.18%)	4427 (20.19%)
4–<5	819 (18.31%)	1110 (15.00%)	711 (15.30%)	623 (14.08%)	169 (15.36%)	3432 (15.61%)
5–<6	346 (7.74%)	638 (8.62%)	401 (8.63%)	515 (11.64%)	77 (7.00%)	1977 (8.73%)
≥6	492 (11.00%)	745 (10.07%)	608 (13.08%)	709 (16.03%)	179 (16.27%)	2733 (13.29%)
Specimens	586	604	616	603	292	2701
EV-positive (%)	420 (71.67%)	328 (54.30%)	311 (50.49%)	227 (37.65%)	153 (52.40%)	1439 (53.28%)
EV-A71(%) ^a^	123 (29.29%)	69 (21.04%)	22 (7.07%)	3 (1.32%)	2 (1.31%)	219 (15.22%)
CVA16 (%) ^b^	106 (25.24%)	50 (15.24%)	130 (41.80%)	112 (49.34%)	16 (10.46%)	414 (28.77%)
Other EVs (%) ^c^	191 (45.48%)	209 (63.72%)	159 (51.13%)	112 (49.34%)	135 (88.24%)	806 (56.01%)
CVA10 (%) ^d^	13 (6.81%)	0 (0.00%)	3 (1.89%)	14 (12.50%)	1 (0.74%)	31 (3.85%)

Notes: “^a, b, c^” was the proportion of EV-A71, CVA16, and other EVs in positive numbers, respectively. “^d^” was the proportion of CVA10 in other EVs.

**Table 2 viruses-15-00694-t002:** The average evolutionary divergence of nucleotide and amino acid sequences within and between genotypes based on the CVA10 *VP1* coding gene.

Genotypes	A	B	C	D	E	F	G
A	*/*	9.4%	8.0%	9.5%	8.1%	7.5%	9.8%
B	*29.6%*	*7.3%* (2.0%)	5.5%	8.4%	5.1%	5.4%	6.5%
C	*28.7%*	*21.6%*	*4.0%* (1.2%)	8.1%	4.1%	4.8%	5.7%
D	*29.9%*	*24.1%*	*24.3%*	*3.1%* (1.2%)	7.8%	6.5%	7.3%
E	*27.8%*	*21.6%*	*18.3%*	*23.3%*	*7.3%* (3.5%)	3.6%	6.6%
F	*27.9%*	*21.7%*	*20.5%*	*21.8%*	*21.0%*	*8.6%* (2.1%)	6.0%
G	*30.5%*	*19.1%*	*17.5%*	*24.6%*	*19.3%*	*20.5%*	*/*

Note: The left-lower data in italics are nucleotide diversity. The right-upper data in normal font are amino acid sequence diversity. Genotype F is based on the available partial *VP1* gene of CVA10 from GenBank, the others used the complete *VP1* region.

## Data Availability

The datasets used and/or analysed during the current study are available from the corresponding author on reasonable request. All the CVA10-Taiyuan sequences obtained during this study were submitted in GenBank under accession numbers OP244619–OP244649.

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
