# Peer review of "Epidemiology of Hand, Foot, and Mouth Disease and Genetic Evolutionary Characteristics of Coxsackievirus A10 in Taiyuan City, Shanxi Province from 2016 to 2020"

_viruses, 2023, doi:10.3390/v15030694_

Round 1

Reviewer 1 Report

please refer to attached file

Reviewer 2 Report

The authors reported on a epidemiology of HFMD in Taiyuan City, Shanxi Province, especially focusing on CVA10.   They showed that CVA10 is one of the important viruses of other EVs apart from EV-A71 and CVA16.   Their phylogenetic analysis indicated that most strains belonged to subtype C2, in which the predominant type changed from C2a in 2016 to C2b and C2c between 2018 and 2019.     My comments are as follows.

1)    (Background 1st paragraph): Small letters such as “ev, cva2, and hfmd” should be changed into capital letters as other parts of the manuscript.

2)     (lines 51-55): The authors described that more research is required, as CVA10 is the new leading cause of severe HFMD.   Did the authors find any relationships between the severity and subtype or something else in this study?

3)    (line 78): Where did the throat swab specimen come from?   One or several hospitals in Taiyuan city?

4)    (Table 1): I would appreciate if you could show not only the numbers in each year but also the total numbers of 2016-2020.   Were there any relationship between the age and serotype of viruses?

5)    (lines 164-169): What kind of enteroviruses were included in “other EVs” apart from CVA10?

6)    (lines 210-211): The authors described that studies focused mostly on the EV-A71 and CVA16 and information about other EVs was still limited.   Can the authors refer any papers related to EV-A71 and CVA16 in Shanxi Province?

7)    (lines 211-213): Did Taiyuan City or Shanxi Province introduced the monovalent EV-A71 vaccine?   If so, how the vaccine has been used (vaccination age, vaccination rate…) ?

8)    (lines 217-241): The authors described that CVA10-associated HFMD outbreaks were reported in many provinces of China.   Are there any differences for the epidemiological characteristics of CVA10-related HFND between Taiyuan City, Shanxi Province and other area?

9)    (Conclusions): The authors described that the surveillance is important to provide a scientific basis for HFMD prevention, control and vaccine development.   If so, not only phylogenetic analysis but also antigenic analysis is important.   Did the authors confirm the antigenic differences among different subtypes?
